# Feasibility of Vitamin C in the Treatment of Post Viral Fatigue with Focus on Long COVID, Based on a Systematic Review of IV Vitamin C on Fatigue

**DOI:** 10.3390/nu13041154

**Published:** 2021-03-31

**Authors:** Claudia Vollbracht, Karin Kraft

**Affiliations:** 1Medical Science Department, Pascoe Pharmazeutische Präparate GmbH, 35383 Giessen, Germany; 2Department of Internal Medicine, University Medicine Rostock, 18057 Rostock, Germany; karin.kraft@med.uni-rostock.de

**Keywords:** ascorbic acid, post-viral fatigue, lack of concentration, sleep disturbances, depression

## Abstract

Fatigue is common not only in cancer patients but also after viral and other infections. Effective treatment options are still very rare. Therefore, the present knowledge on the pathophysiology of fatigue and the potential positive impact of treatment with vitamin C is illustrated. Additionally, the effectiveness of high-dose IV vitamin C in fatigue resulting from various diseases was assessed by a systematic literature review in order to assess the feasibility of vitamin C in post-viral, especially in long COVID, fatigue. Nine clinical studies with 720 participants were identified. Three of the four controlled trials observed a significant decrease in fatigue scores in the vitamin C group compared to the control group. Four of the five observational or before-and-after studies observed a significant reduction in pre–post levels of fatigue. Attendant symptoms of fatigue such as sleep disturbances, lack of concentration, depression, and pain were also frequently alleviated. Oxidative stress, inflammation, and circulatory disorders, which are important contributors to fatigue, are also discussed in long COVID fatigue. Thus, the antioxidant, anti-inflammatory, endothelial-restoring, and immunomodulatory effects of high-dose IV vitamin C might be a suitable treatment option.

## 1. Introduction

Fatigue often occurs as a symptom of severe diseases, such as cancer or autoimmune diseases. Chronic fatigue syndrome (CFS) is defined as a separate clinical entity, although the symptoms are very similar: besides intense fatigue, most patients with CFS report attendant symptoms such as pain, cognitive dysfunction, and unrefreshing sleep [1,2]. Since fatigue is still difficult to treat, there is an urgent need for effective treatment options.

Fatigue is also currently coming into focus as a major symptom of long COVID. Patient data from all over the world show that COVID-19 not only attacks people’s health during the acute infection but also often results in post-infection problems, which are summarized under the term long COVID [3]. SARS-CoV-2 positive persons can be grouped into asymptomatic infection (no symptoms that are consistent with COVID-19), mild or moderate (symptoms but SpO_2_ ≥ 94%), severe (SpO_2_ < 94%, PaO_2_/FiO_2_ < 300 mm Hg, respiratory frequency > 30 breaths/min, or lung infiltrates > 50%), or critical illness (respiratory failure, septic shock, and/or multiple organ dysfunction) [4]. Symptoms of long COVID can overlap with the post–intensive care syndrome that has been described in patients without COVID-19, but symptoms after COVID-19 have also been reported in patients with milder illness, including outpatients. Obviously, COVID-19 is a multi-system disease characterized by organ and vessel dysfunctions mainly caused by cytokine storm and microembolism. Presumably, this also applies to the post-acute recovery phase. The frequency, nature, and causes of long COVID are currently being investigated intensely.

Until the end of 2020, post-viral fatigue syndrome was listed under the WHO indication code G93.3 (CFS). As of January 2021, the WHO has defined new ICD-10 code numbers as part of the attention to COVID-19: U08.9 Personal history of COVID-19, unspecified; and U09.9 Post-COVID-19 condition. 

Post-viral fatigue is associated with various infectious diseases (SARS coronavirus, Epstein–Barr virus, Ross River virus, enteroviruses, human herpesvirus-6, Ebola virus, West Nile virus, Dengue virus, and parvovirus; bacteria such as *Borrelia burgdorferi*, *Coxiella burnetii*, and *Mycoplasma pneumoniae*; and even parasites, such as *Giardia lamblia*), which often show very different symptoms during the acute stage [1]. Post-viral fatigue syndrome is rather similar to CFS. In this context, it is interesting to note that CFS often begins with an infection during a period of increased physical activity or stress. This corresponds to the current situation in which patients with long COVID are or were affected not only by the infection but likely also by psychological and/or somatic stress during the lockdown.

In a recently published cohort study from Wuhan, which investigated 1733 patients after hospitalization for COVID-19 (half of them were younger than 57 years), even 6 months after the acute infection, 63% of those who had recovered suffered from fatigue or muscle weakness, 26% suffered from sleep disturbances, and 23% suffered from anxiety or depression. Patients who had been more severely ill during their hospital stay had more severely impaired pulmonary diffusion capacities and abnormal chest imaging manifestations [5]. The US National Institute for Health Research started a dynamic review on persistent COVID-19 symptoms in October 2020 and pointed out that not only hospitalized patients but also those with milder courses can be affected [3]. 

A recent systematic review and meta-analysis identified more than 50 long-term effects of COVID-19, with fatigue, anosmia, pulmonary dysfunction, abnormal chest X-ray/CT, and neurological disorders being the most common. It was estimated that 80% of individuals with a confirmed COVID-19 diagnosis continued to suffer from at least one problem beyond two weeks following acute infection. Most of the symptoms were similar to the symptomatology developed during the acute phase of COVID-19. The five most common symptoms were fatigue (58%), headache (44%), attention deficit disorder (27%), hair loss (25%), and dyspnea (24%) [6].

Although studies are still rather heterogeneous, it is already clear that post-viral fatigue accompanied by sleep disturbances and cognitive deficits is one of the most common complaints of long COVID.

The pathophysiology of COVID-19 is characterized by inflammation and oxidative stress leading to vascular and organ damage, as well as to the suppression of adaptive immune responses [7]. It can be assumed that the post-acute recovery phase is also accompanied by oxidative stress, inflammation, and thus a deficiency of antioxidants such as vitamin C. To date, post-infectious vitamin C plasma levels have not been evaluated. However, a deficiency is most likely since infections are known to be associated with high consumption of vitamin C, and deficiencies in acute infections are frequent [8], especially for patients with pneumonia and COVID-19 [9,10,11,12,13]. 

A clinically relevant vitamin C deficiency is a disease-eliciting condition, as the water-soluble vitamin is one of the body’s most important antioxidants and is involved as a co-factor in more than 150 metabolic functions [14]. The term “vitamin C” encompasses the terms ascorbic acid and ascorbate. The latter is the biologically active form that is oxidized to dehydroascorbate when reactive oxygen species are neutralized. As an enzymatic co-factor, it is particularly important for the synthesis of collagen and carnitine, the bioavailability of tetrahydrobiopterin, and thus the formation of serotonin, dopamine, and nitric oxide, the synthesis of noradrenaline, the biosynthesis of amidated peptides, the degradation of the transcription factor HIF-1 α, and the hypomethylation of DNA [8,15]. Fatigue, pain, cognitive disorders, and depression-like symptoms are known symptoms of a vitamin C deficiency [16].

It is therefore clinically plausible that vitamin C administration could alleviate fatigue (1) by treating vitamin C deficiency symptoms, including fatigue, and (2) by neuroprotective and vasoprotective effects due to its antioxidant and anti-inflammatory properties.

The aim of this publication is to provide a feasibility analysis of whether the use of intravenous (IV) vitamin C in post-viral fatigue, particularly after COVID-19, should be further investigated. For this purpose, the pathophysiological factors underlying fatigue were investigated through a narrative review, and possible approaches for a therapeutic benefit of vitamin C in this condition were elicited. In addition, a systematic review was conducted to evaluate the study evidence on IV vitamin C in fatigue. The review focuses on high-dose IV vitamin C because, in contrast to oral application, only the IV route results in pharmacological plasma levels (>220 µM) [17,18]. Moreover, the high plasma levels reached after IV application offer the advantage of rapid bioavailability in the tissues [19]. Studies with oral vitamin C administration are often of low quality because vitamin C blood levels were rarely determined, and therefore, bioavailability and compliance could not be verified. This bias can be avoided with IV administration, which has the advantage of 100% bioavailability and compliance and additionally facilitates the circumvention of genetically determined resorption differences, which are described for the vitamin C transporter in patients with COVID-19 [20].

## 2. Materials and Methods

For the narrative feasibility analysis, the search terms “fatigue” and “review” as well as “fatigue” and “oxidative stress” were used in the Medline database. 

For the systematic review, the Medline and Cochrane Central databases were searched: Medline with the mesh terms “fatigue” and “ascorbic acid” and Cochrane Central (PubMed, Embase, ICTRP, CT.gov (accessed on 25 February 2021)) with “fatigue” and “vitamin C”. The results were screened by the authors for clinical studies with IV vitamin C. Eligibility criteria were the evaluation of fatigue by a score and the therapeutic use of IV vitamin C > 1g. Studies detected via secondary literature were supplemented. As fatigue was investigated in the context of the EORTC-Q30 in studies on quality of life in oncological patients treated with high-dose vitamin C, these studies were added to the search result.

## 3. Results

The search in PubMed for the search terms "fatigue" and "ascorbic acid" resulted in 43 publications, the search in Cochrane Central resulted in 62 publications, 6 were identified via publications that used EORTC Q30 questionnaires, and 8 were duplicates. Ninety-three publications were excluded because they were not clinical studies (*n* = 31), were case reports (*n* = 2), or they did not meet the eligibility criteria (because they used oral vitamin C, often combined with several substances (*n* = 37), did not use vitamin C (*n* = 20), or were study registrations without published results (*n* = 3)). From the 10 full-text publications, one was discarded because there was no information as to how intensity of fatigue was measured. (Figure 1).

From the nine identified clinical studies with 720 participants, three were randomized and controlled studies, one was a retrospective controlled cohort study, one was a phase I study, one was a before-and-after study, and the remaining ones were prospective observational studies.

For the evaluation of fatigue, four studies used EORTC QLQ-C30, three used a Likert scale and, two used a numeric rating scale. The IV vitamin C doses administered ranged from appr. 3.5 g to >75 g/day (three studies with >50 g, two studies with 10 g, three studies with 7.5 g, and one with approximately 3.5 g).

Three of the four controlled trials observed a significant decrease in fatigue in the vitamin C group compared to the control group (*p* < 0.005). In all observational before-and-after studies, a reduction in fatigue was reported. In the four studies that performed a statistical comparison of the pre–post values, the differences were significant (*p* < 0.01) (Table 1).

## 4. Discussion

Altogether, nine clinical studies with 720 participants were identified. In three of the four controlled trials, a significant decrease in fatigue was detected in the high-dose vitamin C group compared to the control group. Vitamin C had no effect on acute post-operative fatigue. Four of the five observational or before-and-after studies performed a statistical comparison of pre–post values and observed a significant reduction in fatigue. To date, the effect of IV vitamin C on fatigue has been studied mainly in cancer patients. Additionally, there is one study in herpes zoster [26], one in allergies [27], one post-operative [28], and one in apparently healthy full-time employees [29]. 

Despite the different underlying diseases, high-dose vitamin C showed a significant reduction in fatigue in almost all studies. The most recent study in patients with advanced lung cancer [21] is particularly compelling: while fatigue continued to increase in the control group despite the best supportive therapy, it decreased significantly in the group with vitamin C plus hyperthermia. The oncology studies mostly used the EORCT QOL-C30, which also examines physical and cognitive dysfunction, dyspnea, insomnia, and pain. These complaints were also frequently alleviated by vitamin C. In cancer, very high doses of vitamin C are tested because of its chemotherapeutic potential. Three of the five oncology studies used doses >50 g [21,22,23]. In two studies, the dose was calculated based on body weight (bw) and ranged between 0.8 and 3 g vitamin C per kg bw [22]. For a 75 kg person, this means between 60 and 225 g of vitamin C per infusion. The two remaining studies used much lower (by a factor of 10) doses per application, yet fatigue was significantly reduced [24,25]. This means that very high doses do not seem to be necessary for improving quality of life such as reducing fatigue. In their review of vitamin C in cancer-associated fatigue, Carr et al. [30] discussed the underlying mechanisms of action and concluded that the rapid correction of deficiency states, the effect as a co-factor of enzymatic reactions, and the anti-oxidative and anti-inflammatory effects are particularly important. All these effects do not require extremely high doses of vitamin C. The only study that investigated the effects of IV vitamin C in a viral disease (herpes zoster) also used a smaller amount (7.5 g) but with a high frequency (every second or third day) [26]. Fatigue improved in 78.2%, and impaired concentration improved in 81.8% of the patients. The same dose was used for the treatment of allergies, where fatigue is also a problem that affects the quality of life [27].

While the change in fatigue was only evaluated after 3 or more weeks in most studies, the study in apparently healthy full-time workers [29] reported an acute reduction in fatigue. One of the oncological studies [24] evaluated fatigue after one week and detected significant relief after this short treatment period.

The narrative feasibility analysis revealed that fatigue is most common in autoimmune diseases, intestinal bowel diseases, neurological diseases, and cancer [31,32,33,34]. Shared features of these diseases are inflammation and oxidative stress, which reinforce each other. Oxidative stress seems to be not only a convincing contributor but also a promising biomarker of the treatment of fatigue [35,36,37,38,39]. In cancer patients, an exercise intervention upon cessation of radiation or chemotherapy resulted in a reduction in fatigue [35]. The improvement was accompanied by a significant decrease in markers of oxidative stress. Changes in total and affective fatigue exhibited significant correlations with changes in plasma 8-hydroxy-deoxyguanosine over time, while behavioral and sensory fatigue changes were significantly correlated with protein carbonyls. Increases in antioxidant capacity were significantly correlated with reductions in affective, sensory, and cognitive fatigue [35]. Fatigue in patients with systemic lupus erythematosus with low disease activity is associated with increased markers of oxidative stress (F(2)-isoprostane). In a multivariate model, F(2)-isoprostane was a significant predictor of fatigue severity after adjustment for age, body mass index, pain, and depression [39]. Oxidative stress, impaired sleep homeostasis, mitochondrial dysfunction, immune activation, and (neuro-)inflammation can aggravate each other in a vicious pathophysiological loop in CFS [37]. Even in the pathophysiology of idiopathic CFS, oxidative stress seems to be a key contributor [36]. Compared to healthy controls, patients with idiopathic CFS have significantly elevated markers of oxidative stress (including reactive oxygen species, malondialdehyde, and F2-isoprostane) and reduced levels of antioxidant parameters, which include total antioxidant activity and catalase, superoxide dismutase, SOD, and GSH activity [36]. 

Fatigue is also very well known in cancer: not only does it accompany chemotherapy and radiation, which contribute to oxidative stress, but it can also persist long after completion of oncological treatment [34,40]. 

Inflammation and oxidative stress interfere with neurotransmitter metabolism, resulting in increased glutamatergic and decreased monoaminergic neurotransmission (serotonin, noradrenaline, and dopamine) via differing routes. These negatively affect neurotransmitter functioning in various cerebral areas that are involved in fatigue [41]. In this context, it is important to consider that oxidative stress not only reduces the bioavailability of neurotransmitters due to increased degradation, decreased formation, and distribution but also results in a decrease in antioxidants. Vitamin C is one of the most important endogenous antioxidants and is reduced in many chronic inflammatory diseases such as rheumatoid arthritis, inflammatory bowel diseases, and cancer [42,43,44,45]. Furthermore, together with vitamin C, vitamin B6, B12, and folic acid are important enzymatic cofactors of the synthesis of serotonin, dopamine, and noradrenaline.

Oxidative stress is also a major influencing factor for endothelial dysfunction and circulatory disorders. High-dose IV vitamin C combats overwhelming oxidative stress and restores endothelial and organ function [46]. In the case of COVID-19, oxidative stress not only triggers organ damage but also causes immune thrombosis via the formation of neutrophil extracellular traps (NETs). This results in embolisms and dysfunction of microcirculation [7,47]. The situation is aggravated by the fact that SARS-CoV-2 also penetrates endothelial cells via ACE receptors and triggers a chain reaction of endothelial damage, infiltration of neutrophils, and resulting NETs [48]. As for vitamin C, it is essential for the phagocytosis of consumed neutrophils by macrophages. If this clearance does not take place, necrosis of the neutrophils occurs, leading to NETs and thus to circulatory disorders [8,47]. Therefore, an early application of high-dose vitamin C is proposed to possibly prevent the development of severe COVID-19 courses [7,47]. Indeed, in a first pilot study in COVID-19 patients requiring intensive care, high-dose IV vitamin C significantly improved oxygenation, reduced organ-damaging cytokine storm (IL-6), and showed a trend towards reduced mortality in severely ill patients [49]. A significant reduction in mortality and improvement of oxygen status by high-dose vitamin C was observed in a recent retrospective cohort study [50]. From these findings, it can be hypothesized that vitamin C administration may also be associated with a therapeutic post-viral benefit in the case of persistent symptoms. Randomized controlled trials, such as LOVIT-COVID (NCT04401150) or EVICT-CORONA-ALI (NCT04344184), are still ongoing. 

A recent review of long COVID described abnormal chest X-rays/CT in 34% of the patients 6 months after infection. Markers reported to be elevated were D-dimer, NT-proBNP, C-reactive protein, serum ferritin, procalcitonin, and IL-6 [6], which implies involvement in circulatory disorders, cardiac insufficiency, and inflammatory reactions. 

Another cause of persistent symptoms could be the induction of immune responses to self-epitopes during acute severe COVID-19. First observations point to IgG autoantibodies that are widely associated with myopathies, vasculitis, and antiphospholipid syndromes in SARS-CoV-2 infected subjects [51]. The observation of autoimmune antibodies is interesting, as fatigue is a known major problem in autoimmune diseases such as multiple sclerosis [37,52], rheumatoid arthritis [33,53], diabetes mellitus type 1 [54], systemic lupus erythematosus [39], and inflammatory bowel diseases [55]. 

Inflammation results in an overlap of fatigue, disturbed sleep, cognitive deficits, pain, and depression-like symptoms [41], the very pattern of symptoms observed in long COVID. These factors, which accompany and probably promote fatigue in long COVID, were alleviated in the clinical studies on IV vitamin C.

Therefore, the effects of IV vitamin C on post-viral COVID-19 fatigue should be investigated in clinical trials.

## 5. Conclusions

Oxidative stress and inflammation can cause and maintain fatigue, cognitive impairment, depression, and sleep disturbances. They disrupt the formation and functioning of important neurotransmitters and of blood circulation. Vitamin C is one of the most effective physiological antioxidants, showing anti-inflammatory effects, especially if applied intravenously in pharmacological doses. It restores endothelial function, and it is an enzymatic co-factor in the synthesis of various neurotransmitters.

High-dose IV vitamin C has been investigated in four controlled and five observational or before-and-after studies in patients with cancer, allergies, and herpes zoster infections. The results show a reduction in fatigue and attendant symptoms such as sleep disturbances, depressive symptoms, pain, and cognitive disorders.

COVID-19 is a multisystem disease in which oxidative stress is partly responsible for excessive inflammation and circulatory disorders such as immune thrombosis. Vitamin C deficiency has been demonstrated in COVID-19 and other acute severe infections and should also be investigated in long COVID. Furthermore, the effects of high-dose IV vitamin C on long COVID-associated fatigue should be investigated in clinical trials.

## Figures and Tables

**Figure 1 nutrients-13-01154-f001:**
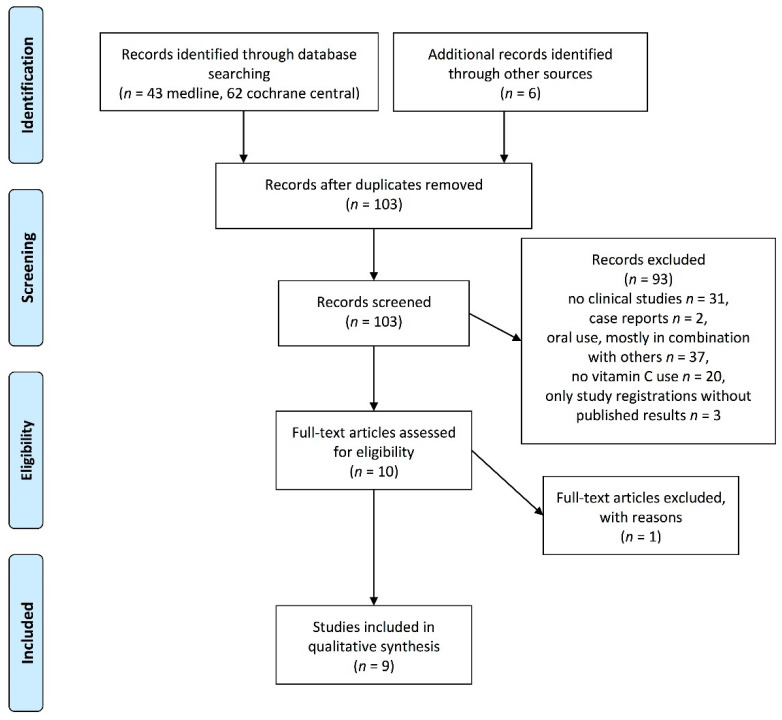
Documentation of study selection for the systematic review according to PRISMA guidelines.

**Table 1 nutrients-13-01154-t001:** Clinical studies investigating intravenous vitamin C in conditions with fatigue. * *p*-value for pre vs. post; ** *p*-value for verum vs. control; bw: body weight; NRS: numeric rating scale.

Reference	Study Type; Number of Patients (n); Underlying Disease	IV Vitamin C Dose	Additional Interventions	Estimation of Fatigue	Impact on Fatigue and Related Parameters
*Oncology*					
[21]	Single-center, phase II, randomized clinical trial; *n* = 97; extensively pretreated patients with advanced, refractory non-small-cell lung cancer	1 g/kg bw, 3 times/week, 25 treatments in total	Vitamin C group received concurrently modulated electro-hyperthermia; both groups received best supportive care	EORTC QLQ-C30	Fatigue (mean ± SD) Verum group: pre: 46.48 ± 17.52, post: 20.63 ± 18.14 (* *p* < 0.0001) Control group: pre: 39.93 ± 20.59, post: 61.34 ± 25.32 (* *p* < 0.0001) (** *p*< 0.0001) Physical function ↑ (** *p* < 0.0001) Cognitive function (** *p* = 0.1026) Dyspnea ↓ (** *p* < 0.0001) Insomnia (** *p* = 0.0772 Pain ↓(*p*** *p* < 0.0001)
[22]	Single-center phase I clinical trial; *n* = 17; patients with refractory, advanced solid tumors (stage III-IV; colon, pancreas, breast, etc.)	0.8–3 g/kg bw, 4 times/week for 4 weeks	None	EORTC QLQ-C30	Fatigue ↓ (pre: 49/ post 11) Physical function ↑ (pre 69/post 87) Cognitive function ↑ (pre 75/post 83) Dyspnea ↓ (pre 24/post 0) Insomnia ↓ (pre 31/post 17) Pain ↓ (pre 36/ post 0)
[23]	Multi-center, prospective observational trial; *n* = 60; patients with advanced tumors (lung, breast, stomach, colonm etc.)	Increasing dosages up to 50 g and more to achieve plasma levels of 350–400 mg/dL 2 times/week for 4 weeks	+/− chemotherapy	EORTC QLQ-C30	Fatigue (mean ± SD) Pre: 42.4 ± 28.7 post: 28.4 25.7 (* *p* < 0.01) Physical function ↑ (* *p* < 0.05) Cognitive function ↑ (* *p* < 0.01) Dyspnea (not significant) Insomnia ↓ (* *p* < 0.01) Pain ↓ (* *p* < 0.05)
[24]	Single-center, prospective before-and-after study; *n* = 39, terminal cancer patients (stomach, colon, lungs, breast, gall bladder, etc.)	10 g 2 times/week for one week	None	EORTC QLQ-C30	Fatigue (mean ± SD) Pre: 52 ± 24, post: 40 ± 19 (* *p* = 0.001) Physical function ↑ (* *p* = 0.037) Cognitive function ↑ (* *p* = 0.002) Dyspnea (*p* = 0.051) Insomnia ↓ (* *p* = 0.029) Pain ↓ (* *p* = 0.013)
[25]	Multi-center, retrospective, cohort study; *n* = 125, patients with breast cancer UICC IIa-IIIb	≥7.5 g at least 1 time/week for at least 4 weeks	+/− chemotherapy, radiation	3-point Likert scale	Fatigue (mean ± SD) During adjuvant therapy (first 6 months after operation): Verum: pre: 1.53 ± 1.11, post: 0.71 ± 0.89 Control: pre 1.68 ± 1.004, post: 1.24 ± 0.936 (** *p* = 0.004) During after care (6–12 month after operation): Verum: 0.34 ± 0.58 Control: 0.64 ± 0.718 (** *p* = 0.023) Sleep disorders ↓ (** *p* = 0.005) Depression ↓ (** *p* = 0.01)
*Infection, allergies*					
[26]	Multi-center, prospective observational trial; *n* = 67; patients with herpes zoster infection	7.5 or 15 g; on average 8 infusions within 2–3 weeks	55.8% received anti-infective drug	4-point Likert scale	Fatigue improved in 78.2% of the patients; Impaired concentration improved in 81.8% of the patients
[27]	Multi-center, prospective observational trial; *n* = 71; patients with respiratory and cutaneous allergies	7.5 g; 2–3 times/week for 2–3 weeks in acute and 11–12 weeks in chronic states	35 % received anti-allergic drugs	4-point Likert scale	Sum score (0–12) of the 4 symptoms: fatigue, sleep disorders, depression, and lack of mental concentration decreased from 5.93 to 1.09 (* *p* < 0.0001) Fatigue improved in 93.5% of patients Sleep disorders improved in 92.5%, depression in 95.5%, and impaired concentration in 91.7%
*Others*					
[28]	Single-center, randomized, double-blind, controlled clinical trial; *n* = 97; patients under-going laparoscopic colectomy	50 mg/ kg bw; Single application after induction of anesthesia	Analgesics	NRS (0–10)	No significant differences in fatigue score 2, 6, and 24 h post operation Pain ↓ (** *p* < 0.05)
[29]	Multi-center, randomized, double-blind, controlled clinical trial; *n* = 147; apparently healthy full-time worker	10 g, single application	None	NRS (0–10)	Fatigue (mean ± SD) Verum: Pre: 5.64 ± 2.02, after 2 h: 5.10 ± 2.04, after 24 h: 4.97 ± 2.33 Control: Pre: 5.54 ± 2.07, after 2 h: 5.31 ± 2.00, after 24 h: 5.66 ± 2.16 (** *p* = 0.004) Plasma vitamin C increased after 2 h, marker for oxidative stress decreased in the verum group (** *p* < 0.001)

## Data Availability

Not applicable.

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
