# Peer review of "Feasibility of Vitamin C in the Treatment of Post Viral Fatigue with Focus on Long COVID, Based on a Systematic Review of IV Vitamin C on Fatigue"

_nutrients, 2021, doi:10.3390/nu13041154_

Round 1

Reviewer 1 Report

A broad review investigating the use of vitamin C treatment in fatigue, and then applying the results of this review to viral-induced fatigue and LONG COVID. The main limitation of this study is its design, as most of the results of this review regard fatigue related to oncological problems more than post-viral fatigue. Before being published various changes should be made.

You should either change the title(and the study ) in a feasibility article on the use of vitamin C in the treatment of LONG COVID induced fatigue, incorporating the systematic review as a paragraph of the article, or keep the article as a systematic review on intralesional vitamin C and fatigue, removing LONG Covid from the discussion.

In both cases, the article needs various changes before publication.

In the introduction section, a small description of vitamin C and its functions would be a great addition to the paper; here an article you could consider: doi: 10.1007/s13668-020-00322-4.

Author Response

Dear Reviewer

Thank you very much for your comments, which we have been pleased to implement. Please find our answers inserted in blue font.

A broad review investigating the use of vitamin C treatment in fatigue, and then applying the results of this review to viral-induced fatigue and LONG COVID. The main limitation of this study is its design, as most of the results of this review regard fatigue related to oncological problems more than post-viral fatigue. Before being published various changes should be made.

You should either change the title(and the study ) in a feasibility article on the use of vitamin C in the treatment of LONG COVID induced fatigue, incorporating the systematic review as a paragraph of the article, or keep the article as a systematic review on intralesional vitamin C and fatigue, removing LONG Covid from the discussion.

Thank you very much for the constructive suggestion. The intention of our publication is indeed a feasibility analysis of vitamin C in post-viral fatigue, which is increasingly observed after COVID-19 disease. The systemic review on vitamin C i.v. in fatigue is intended to show the study evidence in. Your recommendation to clarify this is very important. In addition, we conducted a narrative literature search to find out which pathophysiological mechanisms influence fatigue independently of the underlying disease. And these are mainly inflammation and oxidative stress. Factors that are not only present in autoimmune and cancer diseases, but also in infections and probably even post-viral. This is the clinical rationale or consideration why we conclude that intravenous vitamin C should be investigated in post-viral fatigue. Therefore, we made several changes to the manuscript and changed the title and the abstract accordingly. The main changes regarding the following lines: 14-28; 114-119; 130-6; 150-163; 300-3; 324-341.

In both cases, the article needs various changes before publication.

In the introduction section, a small description of vitamin C and its functions would be a great addition to the paper; here an article you could consider: doi: 10.1007/s13668-020-00322-4.

We took up the suggestion and expanded the description of vitamin C in lines 116 to 123.

Thank you for the constructive criticism, which has greatly improved the publication. We hope that we have realised everything to your satisfaction.

Best regards

Claudia Vollbracht

Reviewer 2 Report

This is a generally well written review exploring the connection between long COVID syndrome and post-viral fatigue, previous studies investigating amelioration of fatigue by IV vitamin C administration, and the potential to apply this to long COVID-related fatigue. The following comments could be considered by the authors in order to further improve the manuscript:

Lines 19, 128, 134 and 213 (and Figure 2 tables). These are not ‘observational’ studies. Observational studies are when the investigators do not intervene with any therapies, but simply observe correlations/associations. These are still intervention studies, just not necessarily comprising a control group ie may be ‘before and after’ study designs, etc.

Line 39. Should clarify differences between SARS-CoV2 infection, COVID-19 and severe/critical COVID-19, i.e. are long-term symptoms more likely after severe COVID-19 (as is observed in post-ICU/sepsis syndromes)?

Lines 59-60. Do the symptoms of long COVID also resemble vitamin C deficiency? They appear to. It is known that vitamin C is severely depleted during viral infections (and COVID-19). I see this is covered in lines 81-87, but may want to mention this a bit earlier and refer to subsequent paragraph. Up to you (may not be necessary).

Lines 157-158. Oxidative stress a promising biomarker of what – fatigue? Or a biomarker of the treatment of fatigue? It can’t really be a biomarker of fatigue as such as it is not sufficiently specific.

Line 194. I believe there are some clinical trials currently running eg LOVIT-COVID that are assessing long term quality of life effects of short term IV vitC administration in severe COVID. May not be specifically assessing fatigue though, but general QoL.

Figure 2 should really be formatted as a table and maybe also presented in landscape as the text font is way too small to be able to read. I think the references in this table may not be correct ie should be formatted with the rest of the paper to ensure the reference numbers are correct. I also think that there should be more discussion of the individual studies, not just the summary data. Maybe the first 3 paragraphs in the discussion section could be moved to the results section (as they are really results) and discussed in a bit more detail.

Conclusions: So should only vitamin C deficiency be investigated in long COVID, or also IV vitamin C administration in those suffering from long COVID-associated fatigue?

Minor comments (mostly grammatical):

Line 34: Replace ‘Especially fatigue…’ with Since fatigue… or Especially since fatigue…

Line 69: Replace ‘suffer of at least one…’ with suffer from at least one…

Line 80: Replace ‘antioxidants as vitamin C.’ with antioxidants such as vitamin C.

Lines 89 and 94 and 193: Replace ‘vitamin C substitution’ with vitamin C administration

Line 89-90 – move 1) to after alleviate fatigue so that sentence makes better grammatical sense

Line 118: replace ‘were no clinical studies’ with were not clinical studies

Line 118-120: maybe put (n=x) instead of just (number) to distinguish from reference numbers

Line 121: replace ‘no information how’ with no information as to how

Line 130: replace ‘vitamin C doses applied’ with vitamin C doses administered

Lines 130-132: replace 3,5 with 3.5 and leave space between numbers and units (g)

Line 144: replace So far, with To date, and replace ‘was studied’ with has been studied

Lines 144-154 – add references to papers being discussed. PMID: 25360419 is another relevant paper to consider.

Line 152: replace ‘impaired concentration in’ with impaired concentration improve in – to improve grammatical sense

Line 157: replace ‘seems not to be only’ with seems to be not only

Line 164: This sentence ‘Disorders of the neurotransmitter metabolism seem to be one main cause.’ is a little vague and confusing.

Line 177: replace ‘In case of COVID-19’ with In the case of COVID-19

Line 199 and 203: these two short paragraphs could probably be combined as discussing same topic.

Line 208: replace ‘symptoms we observe in’ with symptoms observed in (as it is not the authors observing it)

Line 212: delete This

Author Response

Dear Reviewer

Thank you very much for your comments, which we have been pleased to implement. Please find our answers inserted in blue font.

This is a generally well written review exploring the connection between long COVID syndrome and post-viral fatigue, previous studies investigating amelioration of fatigue by IV vitamin C administration, and the potential to apply this to long COVID-related fatigue. The following comments could be considered by the authors in order to further improve the manuscript:

Many thanks for the constructive feedback, which we have gladly implemented. Due to changes requested by Reviewer 1, the line numbers have changed. We refer to the new line numbers in the answers.

Lines 19, 128, 134 and 213 (and Figure 2 tables). These are not ‘observational’ studies. Observational studies are when the investigators do not intervene with any therapies, but simply observe correlations/associations. These are still intervention studies, just not necessarily comprising a control group ie may be ‘before and after’ study designs, etc.

The difference between an observational study (non-interventional, i.e. the drug was used electively by the doctor and the course was documented) and a before-after study (intervention in which the doctor used the drug intentionally) is not always easy to recognise from the publication. We have checked the publications again and the study by Yeom et al. is probably a before-after study. For the other studies, the authors explicitly write in the publication that it is an observational study, so we have accepted this as it is. We changed the designation accordingly in lines 30/31, 202, 209, Table 2 (line 213), 223 and 338.

Line 39. Should clarify differences between SARS-CoV2 infection, COVID-19 and severe/critical COVID-19, i.e. are long-term symptoms more likely after severe COVID-19 (as is observed in post-ICU/sepsis syndromes)?

We included a classification of COVID-19 and referred to–intensive care syndrome (line 52 – 59).

Lines 59-60. Do the symptoms of long COVID also resemble vitamin C deficiency? They appear to. It is known that vitamin C is severely depleted during viral infections (and COVID-19). I see this is covered in lines 81-87, but may want to mention this a bit earlier and refer to subsequent paragraph. Up to you (may not be necessary).

Lines 157-158. Oxidative stress a promising biomarker of what – fatigue? Or a biomarker of the treatment of fatigue? It can’t really be a biomarker of fatigue as such as it is not sufficiently specific.

It was used together with antioxidative capacity as a biomarker of treatment of fatigue. We have changed this in line 263.

Line 194. I believe there are some clinical trials currently running eg LOVIT-COVID that are assessing long term quality of life effects of short term IV vitC administration in severe COVID. May not be specifically assessing fatigue though, but general QoL.

Yes, that is important to mention. We have inserted this in line 303/4.

Figure 2 should really be formatted as a table and maybe also presented in landscape as the text font is way too small to be able to read. I think the references in this table may not be correct ie should be formatted with the rest of the paper to ensure the reference numbers are correct. I also think that there should be more discussion of the individual studies, not just the summary data. Maybe the first 3 paragraphs in the discussion section could be moved to the results section (as they are really results) and discussed in a bit more detail.

We have integrated the table into the Word document. We have included the first 3 paragraphs in the results section and discussed the studies in more detail in the discussion section. Due to the changes in the mns, the insertion of references and the references in the table, we have changed the bibliography.

We extended the discussion and included, among other things, the dose and duration of therapy (line 229 – 259).

Conclusions: So should only vitamin C deficiency be investigated in long COVID, or also IV vitamin C administration in those suffering from long COVID-associated fatigue?

 The treatment should also be investigated. We have added this accordingly (line 324/5)

Minor comments (mostly grammatical):

Line 34: Replace ‘Especially fatigue…’ with Since fatigue… or Especially since fatigue…

Done; because of other changes it is now line 47

Line 69: Replace ‘suffer of at least one…’ with suffer from at least one…

Done; because of other changes it is now line 90

Line 80: Replace ‘antioxidants as vitamin C.’ with antioxidants such as vitamin C.

Done; because of other changes it is now line 109

Lines 89 and 94 and 193: Replace ‘vitamin C substitution’ with vitamin C administration

Done; because of other changes it is now line 126, 140, and 301

Line 89-90 – move 1) to after alleviate fatigue so that sentence makes better grammatical sense

Done; because of other changes it is now line 127

Line 118: replace ‘were no clinical studies’ with were not clinical studies

Done; because of other changes it is now line 190

Line 118-120: maybe put (n=x) instead of just (number) to distinguish from reference numbers

Done; because of other changes it is now line 190 - 193

Line 121: replace ‘no information how’ with no information as to how

Done; because of other changes it is now line 194

Line 130: replace ‘vitamin C doses applied’ with vitamin C doses administered

Done; because of other changes it is now line 205

Lines 130-132: replace 3,5 with 3.5 and leave space between numbers and units (g)

Done; because of other changes it is now line 206/7

Line 144: replace So far, with To date, and replace ‘was studied’ with has been studied

Done; because of other changes it is now line 225

Lines 144-154 – add references to papers being discussed. PMID: 25360419 is another relevant paper to consider.

Done. We included the publication of Carr et al. 2014 in Line 244-249. They discuss the underlying mechanisms of action and conclude that the rapid correction of deficiency states, the effect as a co-factor of enzymatic reactions, the anti-oxidative and anti-inflammatory effects are particularly important.

Line 152: replace ‘impaired concentration in’ with impaired concentration improve in – to improve grammatical sense

Done; because of other changes it is now line 253

Line 157: replace ‘seems not to be only’ with seems to be not only

Done; because of other changes it is now line 262

Line 164: This sentence ‘Disorders of the neurotransmitter metabolism seem to be one main cause.’ is a little vague and confusing.

We deleted that sentence and included it partly in the next sentence 270/1.

Line 177: replace ‘In case of COVID-19’ with In the case of COVID-19

Done; because of other changes it is now line 284

Line 199 and 203: these two short paragraphs could probably be combined as discussing same topic.

If you agree, we would like to leave the paragraphs as they are. The second paragraph discusses the formation of autoantibodies in COVID. This is a completely new point of view, which is discussed in the following paragraph:

The recent review of Long COVID observed abnormal chest x-ray/CT in 34% of the patients 6 month after infection. Markers reported to be elevated were D-dimer, NT-proBNP, C-reactive protein, serum ferritin, procalcitonin and IL-6 [5], which implies involvement in circulatory disorders, cardiac insufficiency and inflammatory reactions.

Another cause of persistent symptoms could be the induction of immune responses to self-epitopes during acute, severe COVID-19. First observations point to IgG autoantibodies widely associated with myopathies, vasculitis, and antiphospholipid syndromes in SARS-CoV-2 infected subjects [46].

Line 208: replace ‘symptoms we observe in’ with symptoms observed in (as it is not the authors observing it)

Done; because of other changes it is now line 321

Line 212: delete This

Done; because of other changes it is now line 337

Thank you for the constructive criticism, which has greatly improved the publication. We hope that we have realised everything to your satisfaction.

Best regards

Claudia Vollbracht

Round 2

Reviewer 1 Report

The authors responded to all queries. The article is in my opinion publishable.

Author Response

Thank you for your approval of the manuscript. As reviewer 2 mentioned :

I do not see the point of adding a narrative review on the pathophysiology of fatigue right in front of your systematic review on the effects of vitamin C (maybe the other reviewer suggested it?). I think this would fit better in the introduction - as a rationale for looking at the antioxidant vitamin C (or even in the discussion as an explanation/mechanism for how vitamin C could be improving fatigue). 

We had changed the results section in the course of the conversion to feasibility. However, you had not explicitly requested this. In order to do justice to both reviewers, we would like to integrate the part back into the discussion section, but present it in more detail. You will find the changes in line: 238-260

We hope you agree.

Best reagards

Claudia Vollbracht

Reviewer 2 Report

Thank you for making the suggested changes. Some minor comments regarding the changes below.

The title seems overly long - can this be shortened to make it more concise? 

I do not see the point of adding a narrative review on the pathophysiology of fatigue right in front of your systematic review on the effects of vitamin C (maybe the other reviewer suggested it?). I think this would fit better in the introduction - as a rationale for looking at the antioxidant vitamin C (or even in the discussion as an explanation/mechanism for how vitamin C could be improving fatigue). 

None of the sentences in the following section have references: "Fatigue is most common in autoimmune diseases and cancer. In cancer patients, an 
exercise intervention upon cessation of radiation or chemotherapy resulted in reduction of fatigue. The improvement was accompanied by a significant decrease in markers of oxidative stress. Changes in total and affective fatigue exhibited significant correlations with changes in plasma 8-hydroxy-deoxyguanosine over time, while behavioral and sensory fatigue changes were significantly correlated with protein carbonyls.

Author Response

Thank you for the advice, which we have implemented as follows.

The title seems overly long - can this be shortened to make it more concise? 

We have shortened the title. The title after the first round of review was:

Feasibility considerations of vitamin C in the treatment of post viral fatigue with focus on Long COVID, based on a systematic review of the effectiveness of i.v. vitamin C on fatigue.

We shortened it to:

Feasibility of vitamin C in the treatment of post viral fatigue with focus on Long COVID, based on a systematic review of i.v. vitamin C on fatigue

I do not see the point of adding a narrative review on the pathophysiology of fatigue right in front of your systematic review on the effects of vitamin C (maybe the other reviewer suggested it?). I think this would fit better in the introduction - as a rationale for looking at the antioxidant vitamin C (or even in the discussion as an explanation/mechanism for how vitamin C could be improving fatigue). 

The integration of the narrative review into the results section actually took place in the course of editing Reviewer 1's comments. He/she had recommended focusing on feasibility and seeing the systematic review as a complement. However, reviewer 1 did not explicitly recommend changing the results section. We had previously discussed this part in less detail in the discussion section. We are happy to follow your recommendation and move this part into the discussion section again, but in more detail than in the first manuscript, and we think that this does meet the recommendations of both reviewers.

The changes can be found in the lines: 238-260

None of the sentences in the following section have references: "Fatigue is most common in autoimmune diseases and cancer.

References were included and the occurrence of fatigue was shown more precisely.:

Line 238 – 240: The narrative feasibility analysis revealed that fatigue is most common in autoimmune diseases, intestinal bowel diseases, neurological diseases, and cancer [32-35].

In cancer patients, an exercise intervention upon cessation of radiation or chemotherapy resulted in reduction of fatigue. The improvement was accompanied by a significant decrease in markers of oxidative stress. Changes in total and affective fatigue exhibited significant correlations with changes in plasma 8-hydroxy-deoxyguanosine over time, while behavioral and sensory fatigue changes were significantly correlated with protein carbonyls.

The whole paragraph refers to reference 36; we have inserted it again at the beginning:

Line 244: In cancer patients, an exercise intervention upon cessation of radiation or chemotherapy resulted in reduction of fatigue [36]. The improvement was accompanied by a significant decrease in markers of oxidative stress. Changes in total and affective fatigue exhibited significant correlations with changes in plasma 8-hydroxy-deoxyguanosine over time, while behavioral and sensory fatigue changes were significantly correlated with protein carbonyls. Increases in antioxidant capacity were significantly correlated with reductions in affective, sensory, and cognitive fatigue [36].

Best regards

Claudia Vollbracht